# Polyphenol Release and Antioxidant Activity of the Encapsulated Antioxidant Crude Extract from Cold Brew Spent Coffee Grounds under Simulated Food Processes and an In Vitro Static Gastrointestinal Model

**DOI:** 10.3390/foods12051000

**Published:** 2023-02-27

**Authors:** Onamon Chongsrimsirisakhol, Tantawan Pirak

**Affiliations:** Product Development, Faculty of Agro-Industry, Kasetsart University, Bangkok 10900, Thailand

**Keywords:** antioxidant crude extract from cold brew spent coffee grounds, encapsulation, in vitro static gastrointestinal model, alginate, maltodextrin, inulin

## Abstract

An ionic gelation technique based on an alginate-calcium-based encapsulation process was prepared as the delivery matrix for antioxidant crude extracts from cold brew spent coffee grounds (350 mg/mL). All the encapsulated samples were treated with different simulated food processes, namely pH 3, pH 7, low-temperature long-time (LTLT) pasteurization, and high-temperature short-time (HTST) pasteurization, to evaluate the stability of the encapsulated matrices. The results showed that alginate (2%, *w/v*)/maltodextrin (2%, *w/v*) (CM), and alginate (2%, *w/v*)/inulin (5%, *w/v*) (CI) could enhance encapsulation efficiency (89.76 and 85.78%, respectively) and provide lower swelling behavior after being treated using the simulated food processes. Both CM and CI could control the release of antioxidants during the gastric phase (2.28–3.98 and 2.52–4.00%, respectively) and gradual release in the intestinal phase (6.80–11.78 and 4.16–12.72%, respectively) compared to pure alginate (CA). In addition, pasteurization treatment at pH 7.0 produced the highest accumulated release of total phenolic content (TPC) and antioxidant activity (DPPH) after digestion in the in vitro gastrointestinal system compared to the other simulated food processes. The thermal process resulted in a greater release of compounds from the encapsulated matrix during the gastric phase. On the other hand, the treatment with pH 3.0 resulted in the lowest accumulated release of TPC and DPPH (5.08 and 5.12%, respectively), which indicated phytochemical protection.

## 1. Introduction

Coffee is one of the most-consumed products globally. In the past decade, coffee production based on the cold brewing process has increased markedly because the product is less bitter and has a balanced taste, making it more popular [1]. However, in the production of coffee products, there are large amounts of spent coffee grounds produced from the cold brewing process (CSCG) and these are mostly unutilized. As the CSCG still contain many bioactive compounds such as polyphenol, one form of utilization could be to extract this compound using a green extraction process (ultrasound assist extraction; UAE) to obtain an antioxidant crude extract from CSCG [2]. In general, these bioactive compounds are prone to decrease during food processing and under the harsh conditions of the gastrointestinal system (low pH and highly active digestion enzymes) which could limit their bioactivity or bioavailability [3]. Therefore, it is necessary to identify a process to enhance the stability of bioactive compounds, as well as improve their bioavailability [3,4,5]. Such a process could involve the use of the encapsulation technique to isolate the compound structure from the external environment [6]. Hydrogel-based encapsulation is a prominent technique to preserve and minimize any change in bioactivity in an intense environment, such as in gastrointestinal conditions. Ionic gelatin is simple, low-cost, and requires no need for specialized equipment [4]. In recent years, research on encapsulating natural polyphenol compounds has been reported, including with *Citrus medica* L. [7], Dandelion (*Taraxacum officials* L.) [8], Black Jamun pulp [9], Beetroot *Beta vulgaris* CV. [10], *Junghans regina* L. [5], and Red dragon fruit (*Hylocereus polyrgizus* L.) [11]. However, the antioxidant crude extract from CSCG encapsulated using an ionic gelation (hydrogel) system has not yet been investigated, especially regarding the release behavior under different food pH, food thermal process, and in vitro gastrointestinal conditions.

The ionotropic encapsulation process (hydrogel) could be created by the use of alginate as it become mildly gelatinous upon contact with calcium ions [7]. The hydrogel produced by the alginate calcium base could respond to the surrounding environment as the pH and ionic strength could be designed to control release under gastrointestinal conditions [12]. Alginate is an unbranched polymer consisting of mannuronic and guluronic acids in various sequences [13]. Normally, the hydrogel created using a single encapsulated polymer (alginate) in the ionic gelation process has some limitations. Encapsulation based on pure alginate results in low mechanical strength, low encapsulated efficiency, and bust release [3]. The efficiency could be increased by adding a copolymer that could act as a solid barrier to obstruct diffusion transportation and rapid release from the alginate-based hydrogel matrix structure [12,14]. Some researchers have attempted to address the limitation of pure alginate hydrogel by mixing alginate with other co-polymers, such as starch, carrageenan, and pectin [7], gellan gum [15], chitosan [5], corn starch [16], modified tapioca starch [14], and inulin [17,18]. Commercial maltodextrin has been characterized by the degree of hydrolysis and expressed as a dextrose equivalent (DE). A high DE represents a higher content of reducing sugar while a low DE (10–15) is referred to as resistant maltodextrin. Inulin is a fructo-oligosaccharide (fructose molecule chain) that can act as dietary fiber. As maltodextrin and inulin have high molecular weights with long molecular chain lengths, these properties could be beneficial to the strength of an alginate-based hydrogel structure. Both maltodextrin and inulin are suitable for using as good protection barriers with a high water solubility, neutral taste and low-cost, which would be beneficial for encapsulation [19,20,21,22].

Therefore, in this study, we hypothesized that the synergistic effect of the alginate and co-polymer made using maltodextrin and inulin could be stabilized and sustain the release of antioxidant crude extract from CSCG using ionic gelatin after passing through food processes and gastrointestinal conditions. The main goal of this research was to investigate the release behavior of antioxidant crude extract from CSCG-formed alginate, alginate/maltodextrin, and alginate/inulin under food process and in vitro gastrointestinal conditions. The alginate-based hydrogel was studied in terms of encapsulation efficiently, swelling behavior, and the release of total phenolic content (TPC) and antioxidant activity through DPPH assay, delivered through different simulated food pH levels, thermal treatments, and in vitro static gastrointestinal conditions to understand the release mechanism of the encapsulated polymers.

## 2. Materials and Methods

### 2.1. Preparation of Antioxidant Crude Extracts from CSCG

The antioxidant crude extract from CSCG was prepared using ultrasound-assisted extraction at 50 °C, ethanol 95% ratio 1:20 (*w/v*) for 40 min using thermo-sonication at 40 kHz (D6 series, GT SONIC, China) before evaporation and freeze-drying [2].

### 2.2. Preparation of Alginate, Alginate/Maltodextrin, and Alginate/Inulin Hydrogel Bead Containing Antioxidant Crude Extracts from CSCG

The encapsulated antioxidant crude extract from CSCG was prepared using an ionic gelation process with an alginate-Ca^2+^ base. The ratio and concentration of the antioxidant crude extract and encapsulated materials (alginate, alginate/maltodextrin, and alginate/inulin) are shown in Table 1. The preparation started by adding encapsulated material into antioxidant crude extract from the CSCG solution (350 mg/mL). Then, maltodextrin (2%, *w/v*) or inulin (5%, *w/v*) was added and mixed using a magnetic stirrer for 10 min at room temperature before the alginate (2%, *w/v*) was added into the mixed solution.

Each solution was thoroughly mixed using a magnetic stirrer for 6 h at room temperature (initial solution) before being injected into the calcium lactase (3% *w/v*) through a stainless-steel needle (21 gauge) with a 10 cm distance between the needle and the calcium lactase surface [12]. The microbeads were left to harden in calcium lactase solution for 30 min. Subsequently, the microbeads were washed with distilled water 3 times through a stainless-steel grid and then dried at room temperature (30 °C) for 24 h before storage at 4 °C in an airtight package (air-dried sample).

### 2.3. Microbead Size

The mean diameter of all encapsulated samples was determined using a set of Vernier calipers (39 microbeads/sample). The microbead size was reported as the average diameter ± SD.

### 2.4. Fourier Transform Infrared (FT-IR) Spectroscopic Analysis

The FT-IR spectroscopic analysis of the antioxidant crude extract from CSCG, empty encapsulate (alginate; A, alginate/maltodextrin; M, alginate/inulin; I), and encapsulated antioxidant crude extract from CSCG by alginate (CA), alginate/maltodextrin (CM), and alginate/inulin (CI) was performed according to Stojanovic et al. [23] with minor modifications. Each microbead sample was freeze-dried before being crushed by a mortar, mixed with potassium bromide (KBr), and compressed into pastilles before being analyzed. The infrared spectra of the sample in KBr were recorded between 400 and 400 cm^−1^ in the transmission mode (Spectrum One, Perkin Elmer, Waltham, CT, United States of America (USA)).

### 2.5. Encapsulation Efficiency

The encapsulation efficiency of all encapsulated samples was expressed as the TPC encapsulated in the microbeads. The method followed Ćujić et al. [18] with slight modification. First, the microbeads were mixed with phosphate buffer saline (PBS) 1M at a 1:10 (*w/v*) ratio before sonicating at room temperature for 30 min (to destroy the encapsulating structure), then centrifuged (8000 rpm for 10 min) before collecting the supernatant and then determining the TPC. The encapsulation efficiency was calculated using Equation (1):% Encapsulation efficiency (EE) = (TPC from microbeads/TPC present in initial solution) × 100(1)

Briefly, the TPC was analyzed using the Folin–Ciocalteu assay. The sample was mixed with calcium carbonate (8% *w/v*) and distilled water at a 1:1:10 (*v/v*) ratio before measuring the color development at 765 nm using a spectrophotometer after having left the reaction in darkness for 60 min [2].

### 2.6. Swelling Characteristics of Alginate, Alginate/Maltodextrin, and Alginate/Inulin Hydrogel Beads

All microbead samples were treated with different pH and thermal conditions (simulated food process). The treatments at pH 3.0 and pH 7.0 were prepared to represent acid and natural food pH levels by adding citric buffer (pH 3.0) and Tris chloride buffer (pH 7.0) onto microbead samples at a 1:2.5 (*w/v*) ratio, soaking for 0–120 min before determining the % swelling. The thermal treatment in this study was at 63 °C for 30 min for the low-temperature long-time (LTLT) pasteurization [21] and at 72 °C, for 20 s for the high-temperature short-time (HTST) pasteurization [22], with the microbeads being mixed with distilled water at a 1:2.5 (*w/v*) ratio before applying each thermal process and then determining the % swelling within 0–120 min. The control sample was the microbeads after air-drying without any treatment.

Before determining the % swelling, any excess surface media was removed using filter paper (Whatman No.1) under a vacuum pump (GAST; diaphragm vacuum pump 07061-42, USA) for 5 min and immediately followed by weighing on an electronic balance (dry microbead weight). The swelling characteristic of the microbeads was expressed in two different directions. The positive swelling percentage indicates swelling behavior after treatment while a negative swelling percentage (-) indicates shrinkage behavior after treatment according to Equation (2).
% Swelling = [(W_1_ – W_0_)/W_0_] × 100(2)
where W_1_ and W_0_ represent the dry weight of the microbeads after the simulated food process and the initial weight of the microbeads before the simulated food process, respectively.

The simulated food process microbeads were further tested in three simulated gastrointestinal fluids: a simulated salivary fluid, SSF, (KCl 15.1 mM, KH_2_PO_4_ 3.7 mM, NaHCO_3_ 13.6 mM, MgCl_2_(H_2_O)_6_ 0.15 mM, (NH_4_)_2_CO_3_ 0.06 mM, and CaCl_2_ 1.5 mM); a simulated gastric fluid, SGF, (KCl 6.9 mM, KH_2_PO_4_ 0.9 mM, NaHCO_3_ 25 mM, NaCl 47 mM, MgCl_2_(H_2_O)_6_ 0.12 mM, (NH_4_)2CO3 0.5 mM, and CaCl_2_ 0.15 mM); and a simulated intestinal fluid, SIF, (KCl 6.8 mM, KH_2_PO_4_ 0.8 mM, NaHCO_3_ 85 mM, NaCl 38.4 mM, MgCl_2_(H_2_O)_6_ 0.33 mM, and CaCl_2_ 0.6 mM). All three types were tested at 37°C in a shaking water bath at 200 rpm for 120 min, with any excess surface media being removed using filter paper (Whatman No.1) under a vacuum pump for 5 min and immediately followed by weighing on an electronic balance before determining the % swelling (according to Equation (2), where W_1_ and W_0_ represent, for each of the three simulated gastrointestinal fluids, the dry weight of the simulated food process microbeads after and the initial weight of the microbeads before the simulated food process, respectively). All the simulated gastrointestinal fluid samples were prepared according to Brodkorb et al. [24].

### 2.7. Release Behavior of TPC and Antioxidant Activity (DPPH) from Alginate, Alginate/Maltodextrin, and Alginate/Inulin Hydrogel Beads

All the microbead samples were treated with pH 3.0 (citric buffer) and pH 7.0 (Tris chloride buffer) at a 1:2.5 ratio (*w/v*) for 30 min and 60 min, respectively, before passing through LTLT pasteurization and HTST pasteurization as described in Section 2.6 and then they were analyzed for the released and remaining profiles of the TPC and antioxidant activity (DPPH). The TPC and DPPH analyses were performed according to Chongsrimsirisakhol and Pirak [2]. The antioxidant activity of the sample was analyzed using the DPPH method and quantified by comparing to the Trolox standard curve. The sample was mixed with DPPH solution (0.2 mM) at a 1:3 (*v/v*) ratio and the color development was measured at 517 nm using a spectrophotometer after 30 min of incubation in darkness.

The release behavior was studied under 3 different conditions—simulated food process, simulated gastrointestinal fluid, and simulated gastrointestinal (in vitro digestion).

#### 2.7.1. Released and Remaining TPC and Antioxidant Activity (DPPH) of Microbeads under Simulated Food Process (pH and Thermal Process)

After finishing each simulated food process treatment previously described in the Section 2.6, the samples were centrifuged at 5000 rpm for 10 min before determining the released (supernatant) and remaining (precipitant) parts of the TPC and antioxidant activity. The % released amounts were calculated using Equations (3) and (4):% TPC released = (TPC of supernatant/Initial TPC in encapsulated sample) × 100(3)
% DPPH released = (DPPH of supernatant/Initial DPPH in encapsulated sample) × 100(4)

The initial TPC in the encapsulated sample was defined as the TPC encapsulated in microbeads before the simulated food process treatment. The TPC was determined according to Section 2.5.

The precipitant was mixed with PBS (1M) at a 1:10 (*w/v*) ratio before sonicating (30 min) at room temperature and then centrifuged (8000 rpm for 10 min) before collecting the supernatant and determining the TPC and DPPH, which were calculated as the remaining percentage using Equations (5) and (6):% Remaining TPC = (TPC of precipitant/Initial TPC in encapsulated sample) × 100(5)
% Remaining DPPH = (DPPH of precipitant/Initial DPPH in encapsulated sample) × 100(6)

#### 2.7.2. TPC Release of Microbeads under Simulated Gastrointestinal Fluid (SSF, SGF, and SIF) Conditions

The thermally treated samples from Section 2.6 were further subjected to simulated gastrointestinal fluid conditions according to the INFOGEST standardized method (modified from Brodkorb et al. [24]) in the amount of 3.5 g of microbeads (weighing after removing any excess surface media using filter paper (Whatman No.1) under a vacuum pump). The simulated gastrointestinal fluid condition (without digestive enzyme and bile salt) consisted of three phases: the oral, gastric, and upper part of the intestine. The assay started after incubating the treated microbeads after each simulated food process in the oral phase, where the SSF was mixed with microbead samples for 2 min (M) in a shaking water bath at 200 rpm and 37 °C and then further incubated in SGF (pH adjusted to pH 3.0 using HCl) for 120 min in a shaking water bath at 200 rpm and 37 °C. The digestion phase in the upper part of the small intestine started after incubating gastric chyme for 120 min at pH 7.0 (pH adjusted using NaOH) in a shaking water bath at 200 rpm and 37 °C. During the gastric and intestinal phases, a sample was taken every 30, 60, 90, and 120 min. The collected sample at each simulated gastrointestinal stage was centrifuged at 5000 rpm for 10 min and the supernatant was collected before determining the percentage of released TPC and antioxidant activity (DPPH) using Equations (7) and (8):% TPC released = (TPC of supernatant after simulated gastrointestinal fluid condition/Initial TPC in encapsulated sample) × 100(7)
% DPPH released = (DPPH of supernatant after simulated gastrointestinal fluid condition/Initial DPPH in encapsulated sample) × 100(8)

The initial TPC and DPPH in the encapsulated sample were defined as the TPC and DPPH encapsulated in microbeads before the simulated food process treatment. The TPC and DPPH were evaluated after de-encapsulating the sample using PBS, according to Section 2.2.

#### 2.7.3. TPC Release of Microbeads under Simulated Gastrointestinal Conditions

The simulated gastrointestinal conditions with the digestive enzymes and bile salt (in vitro digestion) proceeded according to the method described in Section 2.7.2; however, 31.62 mg/mL of salivary amylase (95.19 U/mg) was added in the oral phase and 30.78 mg/mL of pepsin (2599.2 U/mg) was added in the gastric phase, while 121.21 mg/mL of pancreatin (6.6 U/mg) and bile salt (153.85 mg/mL) were included in the intestinal phase [24]. The released amounts of TPC and antioxidant activity (DPPH) were calculated at each in vitro digestion stage using Equations (9) and (10):% TPC released = (TPC of supernatant after in vitro digestion/Initial TPC  in encapsulated sample) × 100(9)
% DPPH released = (DPPH of supernatant after in vitro digestion/Initial DPPH  in encapsulated sample) × 100(10)


### 2.8. Statistical Analysis

Each experiment was performed in duplicate with three replications. A full factorial Completely Randomized Design was applied for each parameter in the study. The statistical significance of each variable was determined at the 5% probability level (Duncan’s post hoc test using the SPSS version 19 software).

## 3. Results

### 3.1. Characterization of Microbeads

The microbeads were characterized based on their size, FI-IR spectra, and encapsulation efficiency.

#### 3.1.1. Average Bead Size of Alginate, Alginate/Maltodextrin, and Alginate/Inulin Hydrogel Beads

The mean microbead diameter of CA was higher than for CM and CI with no significant differences among the different encapsulated materials as shown in Table 2. The results were attributed to the structure of CM and CI being denser than CA due to the presence of the co-polymer resulting in less water being present in the hydrogel matrix, resulting in their smaller size [25].

#### 3.1.2. Analysis by FT-IR

This study used FT-IR to identify the functional groups and characterized the chemical compatibility of antioxidant crude extract from CSCG, empty encapsulate microbeads (A, M, and I), and encapsulated materials (CA, CM, and CI) (Figure 1).

The antioxidant crude extract from CSCG has a comparable signature band to the normal polyphenols in its spectra. The antioxidant crude extract from CSCG showed the peak spectrum at 3300 cm^−1^, 1640 cm^−1^, and 1019 cm^−1^ which represented the O-H groups of a phenolic compound, C=C vibration of an aromatic ring, and an alcohol group from polyphenol, respectively (Figure 1A). These findings were similar to Chan et al. [26], Bušić et al. [8], and Ćujić et al. [18]. The presence of the antioxidant crude extract from CSCG in encapsulated microbeads was confirmed by the presence of the band at 1019 cm^−1^ in encapsulated antioxidant crude extract from CSCG which were not observed in empty microbeads spectra.

The FT-IR spectra of encapsulated antioxidant crude extract from CSCG with different encapsulating materials are illustrated in Figure 1B. The FT-IR spectra of CA in KBr revealed a peak band at 3283.89, 1580.75, 1411.47, and 1019.82 cm^−1^, reflective of the O-H stretching vibration, COO- (asymmetric of carboxylate salt), COO- (symmetric of carboxylate salt), and C-O-C stretch in the structure, respectively. These results agreed with those found in the literature [27,28,29]. The peak spectra of CM and CI were very similar to the CA spectrum. However, when compared to the CA and CI, the CM had a more intense spectra peak at 1650 cm^−1^. This characteristic peak corresponds to conjugated C=C or C=O stretching vibrations [26] indicating more C=O or C=C bonding in CM. The stretching peak of the COO- group was less intense in CM and CI when compared to CA. This observation could be caused by the interaction between alginate and the polymer chain of maltodextrin or inulin, as the carboxylic group of alginate was a potential active binging site [23].

#### 3.1.3. Encapsulation Efficiency of Alginate, Alginate/Maltodextrin, and Alginate/Inulin Microbeads

The encapsulation efficiency of the hydrogel (Table 3) mainly depended on phenolic compounds and encapsulated material [6]. In this research, encapsulation efficiency was calculated based on the TPC retained in the encapsulated sample after cross-linking with the Ca^2+^ ion. The results showed that adding maltodextrin and inulin significantly (*p* ≤ 0.05) increased the encapsulation efficiency (Table 3). The % encapsulated efficiently increased from 83.64% to 89.76% and 85.78% for CM and CI, respectively. However, from the preliminary study of the suitable encapsulated material concentrations, there was no significant difference in % encapsulation efficiently observed when the maltodextrin concentration increased from 2 to 3% (*w/v*) nor when the inulin concentration increased from 5 to 10% (*w/v*) (results not shown). These results indicated that the higher co-polymer concentration did not improve the encapsulation efficiency. Consequently, the 2% (*w/v*) maltodextrin and 5% (*w/v*) inulin were selected as the optimal co-polymer concentrations throughout the remaining experiments. Added maltodextrin and inulin could modify the alginate-based hydrogel structure either internally or externally because the polymer completely occupied the interstitial space that impacted the porosity of the hydrogel matrix. Furthermore, the higher affinity of the polar part of the polyphenol interacted with the OH group in the maltodextrin or inulin compared with only a carboxylate group in the pure alginate. These interactions could result in greater stability and a high viscosity, resulting in less polyphenol being lost during the gelation or crosslinking processes [6,17,23,30,31]. The high % encapsulation efficiency in the alginate/inulin hydrogel was also observed by Balanč et al. [17]. However, the % encapsulate efficiency of alginate/inulin reported by Ćujić et al. [18] was not significantly different compared to the use of a pure alginate structure. The difference might be due to the difference in concentrations and methods used between experiments.

### 3.2. Swelling Characteristics of Alginate, Alginate/Maltodextrin, and Alginate/Inulin Microbeads

The swelling characteristic is an important factor that determines the release profile of the encapsulated compound in the alginate-based hydrogel [3,32]. The swelling behavior occurred due to the void region between the polymer network absorbing external water into the hydrogel structure, until reaching the equilibrium state, which happened when the force of cross-linking and the osmotic pressure were equal [12]. The % swelling of all encapsulated samples were negative which indicated the shrinkage of the microbeads during treatment at different pH levels and with or without thermal treatment as shown in Table 4. The % swelling of all encapsulate samples for the pH 3.0 and pH 7.0 solutions decreased and increased, respectively, compared to the control sample. These results could be explained by the protonation of the alginate carboxylate group below the pKa value and the deprotonation above the pKa values, resulting in lower and higher electrostatic repulsion at pH 3.0 and pH 7.0, respectively. The low swelling in a low pH environment was due to a decrease in electrostatic repulsion that led to a polymer–polymer interaction which dominated the polymer–water interaction, creating a compact, dense hydrogel structure that resulted in low swelling or shrinkage [12,32]. Furthermore, the interaction between the calcium-alginate at the core of the microbeads prevented the structure disintegrating [33]. However, in a pH 7.0 environment, the low electrostatic repulsion and the relaxation of the -COO^-^ group were mainly observed, caused by the osmotic pressure inside the hydrogel structure increasing due to the increase in the free H^+^ concentration resulting from the deprotonation process that promoted water uptake and a high swelling percentage [12,25]. The combination with the increase in the electrostatic repulsion between the deprotonated carboxylate groups of the alginate caused a chain relaxation and also enhanced swelling [12].

Heating at 63 °C and 72 °C significantly reduced the encapsulated sample size as a result of water loss due to the external force affecting the hydrogel structure. The higher temperature resulted in a greater size reduction. These findings were similar to Kim et al. [34] who reported a reduction in hydrogel size upon exposure to high temperatures. The swelling behavior of the encapsulated sample treated with a combination of pH and thermal treatment showed a similar swelling degree when treated with the same thermal process combined with the different pH conditions. This could be attributed to the thermal treatment dominating the pH effect.

The swelling behavior of CA was higher than for CM and CI, perhaps as a result of lower diffusive phenomena and higher surface adhesion created by the use of maltodextrin and inulin [32]. Apoora et al. [3] reported that the swelling of a copolymer with a hydrophilic nature depended on the OH and COOH groups. The low concentration of hydrophilic co-polymer in the hydrogel could result in high swelling because the water molecule interacts with the OH and COOH groups. However, if the concentrations were high enough, the swelling would decrease as the tight network formation restarted the swelling process. Furthermore, the presence of the co-polymer provided a denser structure and increased the osmotic pressure resistance. One of the reasons that the presence of maltodextrin and inulin had less of an effect at different pH levels was the lower content of alginate in the hydrogel structure compared to using pure alginate [35]. However, alginate was still the main polymer that dominated the swelling profile [12].

The swelling behavior was explored in SGF (pH 3.0) and SIF (pH 7.0) to identify the adaptability of the encapsulated sample to gastrointestinal conditions. The swelling behaviors of all encapsulated samples in SGF and SIF are shown in Figure 2 and Figure 3. For both the SGF and SIF conditions, the shrinkage and swelling effect of the encapsulated samples treated using the thermal treatment with or without the different pH conditions was not observed compared with the control and the encapsulated samples only treated in different pH conditions. These findings might have been due to the encapsulated hydrogel structure already being compacted and dense after being introduced into the thermal treatment. This compact structure could lower the diffusion of external media, resulting in lower swelling behavior. Hence, the highest swelling was observed in the pH 7.0 treatment sample because this treatment had already caused the hydrogel structure to become swollen, and this effect could then be more easily diffused in the external medium so that a high swelling behavior was observed.

The shrinkage of microbead samples in SGF occurred due to the protonation of the free carbonyl group into the unionized carboxylate group occurring at a pH below the pKa value promoting the formation of an H-bond. This bonding occurred because a decrease in the repulsive charge led to an adjacent alginate polymer chain. The presence of Ca^2+^ ions within the hydrogel structure mostly disappeared, resulting in an alginic acid gel [10,13,36]. In addition, the dissociation of Ca^2+^ ions at low pH resulted in the acid gel due to the formation of -COO^-^ and H^+^ (protonation) allowing alginate molecule structures close to each other due to hydrogen bonding [37]. The swelling under SIF occurred due to the pH environment being above the pKa of alginate and enhancing the completely deprotonated -COOH into its anionic molecular form (-COO^-^) that subsequently increased electrostatic repulsive forces between the -COO^-^ groups [36,38]. Another reason was the presence of monovalent ions (sodium and potassium) in the external solution that could undergo ion exchange with Ca^2+^ that was already bonded with the carboxylate group of the mannuronate sequence (M block), resulting in a higher electrostatic interaction with the free negatively charged carboxyl groups that enhanced swelling and chain relaxation [10,13,37]. During the later stages of swelling, the Ca^2+^ ion bonded with the -COO^-^ of the guluronate sequence (G block) and started to exchange with the Na^+^ ion then, resulting in the egg-box model starting to break. The calcium exchange of the G block was more delayed than for the M block due to the stronger auto-cooperative binding of Ca^2+^ ions at the G block causing the delayed interaction [13]. In addition, the swelling of the alginate-based hydrogel was affected by the presence of phosphate ions, since their higher affinity than that of the Ca^2+^ ions resulted in carboxylate ionization. These factors caused the dissociation of the carboxylate group, leading to a difference in osmotic pressure and then causing solution flux in the hydrogel structure [35]. A similar swelling result in both SGF and SIF has been reported by many researchers [17,30,39]. However, our finding was different from Abd EI-Ghaffar et al. [12] who reported an increase in % swelling for both SGF and SIF. This difference may have been due to the type of alginate used (M:G ratio) and the hydrogel water content.

### 3.3. Release Behavior of TPC and DPPH from Alginate, Alginate/Maltodextrin, and Alginate/Inulin Microbeads

The encapsulation technique was used to protect the sensitive compounds from the harsh external environment until it reached the target release. In general, the release or control of alginate-calcium hydrogel can be manipulated through the different solvents, pH, temperature, and pressure [6]. In food, various processing conditions are normally applied that could damage the activity and content of active compounds; consequently, it is important to investigate the release in these food processes. Thus, both simulated conditions for the food pH and the thermal process were applied using encapsulated antioxidant crude extract from CSCG to understand the effect on the TPC and DPPH release profiles.

#### 3.3.1. Released and Remaining TPC and Antioxidant Activity (DPPH) from Microbeads under Different pH and Thermal Treatments

The compound in the encapsulated samples was immobilized at a temperature below the glass transition temperature. Hydration and relaxation of the polymer chain had to occur to allow the compound to be released [30]. In the current study, the simulated food pH and thermal treatments could be factors affecting the release of compounds from the encapsulated structure. The % release from all encapsulated samples under different pH conditions and thermal treatments are summarized in Table 5 and Table 6. The pH 3.0 and pH 7.0 treatments used in the current study represent food pH conditions for high acid and low acid foods, respectively. The % TPC release levels of CA at both pH 3.0 and pH 7.0 were significantly higher than for CM and CI. These results could be attributed to the presence of the co-polymer as maltodextrin and inulin could act as a solid barrier to obstruct the diffusion of the compound from the hydrogel matrix [40]. The shrinkage in SGF was one of the reasons for the high retention level of the compound in the hydrogel matrix during the strong gastric conditions [36]).

At pH 7.0, the % TPC release was higher than for pH 3.0 in the solution for all encapsulated samples. These observations were related to the findings reported by Alborzi et al. [40] that the structure of the alginate-based hydrogel depends on the surrounding pH environment and specifically whether the pH was below or above the pKa of alginate. In the presence of different pH conditions, two different types of interaction dominated the calcium-alginate-based structure due to the charge repulsion between the dissociated carboxyl group and the ionized carboxyl group. As the pKa of alginate varied in the range 3.3–3.7 (depending on the ratio of the guluronic acid and mannuronic acid groups present in the polymer), if the surrounding environment pH was below the alginate pKa, the alginate structure could be protonated and produce a compact, collapsed network, with low electrostatic repulsion through aggregation, resulting in low swelling, as was previously mentioned in the swelling behavior result. This hydrogel structure could immobilize the encapsulated compound, with lower % TPC and DPPH release being observed [36,40]. In the current research, a low release content (<0.09%) at a pH below the pKa was observed. However, Alborzi et al. [40] reported that the release from the alginate hydrogel structure was around 21%, where this large difference perhaps was contributed to by the difference in the mannuronic acid (M) and guluronic acid (G) ratio used because a lower M:G ratio might create a less stable hydrogel structure.

However, at a pH above the alginate pKa, the alginate structure had increasingly deprotonated negative charge polymers, leading to higher electrostatic repulsion between the polymer chain segments of the alginate, resulting in the compound being more easily released as the structure loosened with high swelling [36,40]. Many researchers have reported that the release under a pH 7.0 buffer was due to the ion exchange of Ca^2+^ ion from the calcium-alginate structure with a Na^+^ ion present in the buffer solution, leading to a higher repulsion force [9]. However, the pH 7.0 buffer solution used in the current experiment was Tris buffer which contained no Na^+^ ions; nonetheless, the release was still observed due to NH^4+^ being exchanged with the Ca^2+^ ion and releasing some TPC and DPPH.

Two thermal treatments (HTST pasteurization and pasteurization) were used to analyze the thermal stability of the encapsulated sample in the current experiment. The encapsulated sample remained stable for both heat treatments, with higher release in pasteurization compared to HTST pasteurization because of the longer explosion time and high temperature under the pasteurization conditions than for HTST pasteurization. Lopez de Dicastillo et al. [41] also reported the release of TPC (10%) from an alginate-based hydrogel under a sterilization process, while a baking process produced approximately 55% release because the baking process used a higher temperature and longer explosion time than the heat treatment; thus, more release was observed. The encapsulated samples treated with combinations of different pH levels with the same thermal treatment resulted in similar levels of TPC and DPPH release compared to the same thermal treatment alone (Table 7 and Table 8). These findings could be explained through the similar swelling behavior shown in the previous results. In the current research, the swelling behavior was highly correlated to the % TPC and DPPH released under the different pH conditions and thermal treatments.

The remaining percentage of the treatment with pasteurization was the lowest when compared to other treatments. In addition, some losses in the TPC and DPPH were observed in all thermal treatments. This was attributed to the thermal treatment degrading and reducing the antioxidant compounds that mainly occurred in the pasteurized encapsulated sample compared to the HTST pasteurized sample due to the longer heating time.

#### 3.3.2. Release of TPC and Antioxidant Activity (DPPH) from Microbeads under Simulated Gastrointestinal Fluid Conditions

The static in vitro gastrointestinal model as described by the INFOGEST international network was applied to investigate the resistance ability of the encapsulated antioxidant crude extracts from CSCG in different encapsulation materials (alginate, alginate/maltodextrin, and alginate/inulin) in terms of the % TPC and DPPH released (Figure 4 and Figure 5).

The gastrointestinal conditions can be separated into three main phases: oral, gastric, and intestinal. All the encapsulated samples had similar TPC and DPPH release behaviors, as the release increased along with the digestion period. In vitro release levels under simulated gastrointestinal fluid were in the ranges 0–3% in SSF, 1.7–7.0% in SGF, and 2.59–18.69% in SIF. There was a higher release in SIF compared to the other simulated digestion phases. The release of TPC and DPPH of all treated encapsulated samples corresponded to the swelling behavior under SGF and SIF. The lower swelling behavior of the encapsulated samples treated using the thermal process produced lower release under SGF and SIF conditions. The more compact and denser hydrogel structure caused by the thermal treatment prevented the compound diffusing from the hydrogel matrix. In contrast, the treated encapsulated sample with pH 7.0 had the highest release profile compared to pH 3.0 and the thermal treatment samples with or without different pH levels. These findings were also related to the swelling behavior in the previous results.

The release of the compound from the hydrophilic gel matrix was controlled by the hydration rate, the content of the compound load in the hydrogel structure, physicochemical properties, formulation, manufacturing process parameters, and the ability of the polymer to re-create the gel structure [42,43]. During the oral phase, there was a low level of release for all the encapsulated samples. This could be attributed to SSF containing Na^+^ ions which could exchange with Ca^2+^ ions in the hydrogel structure with a subsequent low level of release as the interaction time was low (2 min) [5].

After continuing the process into the gastric phase, release increased compared to the oral phase for all the encapsulated samples. The release of TPC was sufficient in the gastric phase with the fastest release at 30 min but remained mostly in equilibrium during the remainder of the gastric phase (30–120 min). Most of the TPC compound remained trapped in the hydrogel structure after the simulated gastric conditions. For the low pH, the remaining or free carboxyl groups after treatment could still be protonated under SGF, resulting in higher electrostatic repulsion and the formation of insoluble alginic acid that caused low swelling and hindered the TPC release in all types of encapsulated samples [11,38,44,45]. Furthermore, another possible reason for the fast release during the gastric phase was the erosion and weakening of the alginate-based structure as the acid was hydrolyzed [14]. The low % TPC and DPPH release in the gastric phase could indicate the stability of all the encapsulated samples exposed to acid conditions before passing through the intestinal phase so that a higher quality and quantity could then be absorbed into the bloodstream and promote health effects [8,43,46,47].

The rapid increase of % TPC and DPPH release during the intestinal phase was observed at the end of the digestion process. In SIF, the increase in pH above the structural pKa value led to deprotonation and the ionization of the remaining free alginate carboxyl groups. As the SIF contained a high concentration of K^+^ and Na^+^ ions, the possibility of these monovalent ions replacing Ca^2+^, which was linked to the -COO^-^ group of the alginate, caused the carboxyl group of the alginate to be relaxed, increasing electrostatic repulsion between the negative charge of the mannuronate sequence, leading to an increased release [8,32,38,45,48]. Another possible explanation for the high release in SIF was the high swelling causing greater exposure of the alginate-based structure to the dissolution medium, creating a larger pore size that contributed to high diffusion [42,49]. As the encapsulated structure still remained intact and was not degraded at the end of the simulated gastrointestinal treatment, it could be assumed that the replacement with monovalent ions of the calcium ion did not proceed into the egg box structure of the guluronate sequence, which would have broken the egg-box model structure [32]. In addition, the presence of a high phosphate ion concentration in SIF resulted in a higher affinity for Ca^2+^ than for the carboxyl group of alginate which induced the disassociation of the calcium-alginate gel matrix [33,44]. In addition to the interaction between polymer-polymer chains contributing to the release profile under SIF, the interaction of the polymer-entrapment compound influences the release profile. The antioxidant crude extract from CSCG contained a small net charge and so did not interact with the carboxylate or hydroxyl groups of the alginate, maltodextrin, and inulin, leading to high diffusion levels [12,50].

The current research added maltodextrin and inulin to the alginate structure to prevent the rapid release of the TPC and DPPH from the alginate’s porous structure. Not surprisingly, the % TPC and DPPH release levels of all CM and CI encapsulated samples were lower than for CA throughout all phases of simulated gastrointestinal conditions. Pure alginate hydrogel provides low mechanical strength, resulting in a rapid release after encounters with a harsh environment. The presence of the co-polymer could promote tortuosity that delayed the internal transport of TPC and sustained the release by physical obstruction and acting as a greater barrier to solvent flow, making diffusion transport more difficult for the compound and causing its release from the hydrogel matrix [8,12,14,30]. The CI-encapsulated samples had a lower level of release compared to the CM-encapsulated samples that was attributed to the higher concentration of inulin present in the hydrogel structure. In addition, the molecular weight of inulin is higher than for maltodextrin, which would benefit preventing compound diffusion. The results were similar to Bušić et al. [8] who reported that a dual copolymer-hydrogel structure could retard release compared to the use of pure alginate in both SGF and SIF. Ćujić et al. [18] and López-Córdoba et al. [30] reported that the level of compound release from the hydrogel network depended on the type of polymer used, the physical state, the matrix structure, and the content of the active compound inside. The difference between the TPC and DPPH release levels for the in vitro gastrointestinal conditions was mainly related to the swelling behavior under both SGF and SIF. Apoorva et al. [3] and Chuang et al. [51], also reported that the release pattern in the gastrointestinal model was related to swelling behavior with lower swelling resulting in a lower release level.

#### 3.3.3. Release of TPC and Antioxidant Activity (DPPH) from Microbeads under Simulated Gastrointestinal Conditions (In vitro Digestion)

The simulated gastrointestinal conditions, including a digestive enzyme were used to better investigate the effects of ionic strength, pH, and the digestive enzyme on alginate, alginate/maltodextrin, and alginate/inulin. In general, the gastrointestinal conditions (pH, enzymes, and strong ionic compounds) could have an effect on the polyphenols and their antioxidant activity [7]. Encapsulation techniques were applied to reduce the degradation of their activity. The primary object of the current research was to develop the process and the encapsulated material to minimize the release and change of antioxidant crude extracts from CSCG under gastrointestinal conditions after passing through the different simulated food pH conditions and thermal processes.

The results showed there were no significant differences between the release levels of TPC and DPPH under both simulated gastrointestinal conditions in the oral phase (with and without amylase enzyme; Figure 6 and Figure 7). These results could suggest that the salivary amylase did not have an impact on the alginate-based hydrogel, even with the added maltodextrin and inulin in the structure, which agreed with the report of Feng et al. [5] who found no release in the oral phase when using only salivary amylase. Ziar et al. [52] also reported that in the presence of salivary amylase at 50 U, the release was not significantly different from their non-enzyme treatment.

In the gastric phase, the steady release was caused by the strong acid and high ionic strength rather than the effect of the pepsin because the release levels were not significantly different between the simulated gastrointestinal conditions with or without pepsin in all encapsulated samples (Figure 8 and Figure 9).

After the intestinal phases, there were high levels of release from all encapsulated samples with no significant differences among treatments with and without bile salt (Figure 10 and Figure 11). Feng et al. [5] reported similar observations in both the gastric and intestinal phases of simulated gastrointestinal conditions. López Córdoba et al. [30] reported that there was no significant difference in TPC release under conditions of nondigestive enzymes and digestive enzymes during the digestion period, perhaps because these enzymes had no impact on phytochemical digestion.

## 4. Conclusions

The current study compared the swelling behavior and % release (TPC and DPPH) of CA, CM, and CI after passing through different food processes and in vitro static gastrointestinal conditions. The co-polymers (maltodextrin and inulin) could sustain the release of encapsulated antioxidant crude extracts from CSCG after passing through simulated food processes and simulated gastrointestinal conditions, thus providing an ideal system for oral delivery. The release of TPC and DPPH with the pH 3.0 treatment was lower than for the pH 7.0 condition, which was correlated with a lower swelling percentage for the former. In addition, the release under LTLT pasteurization was higher than for HTST pasteurization because a more intense thermal treatment was used in the latter process. The release following the combined simulated food process treatment (pH and thermal) was mainly driven by the different thermal treatments, not the pH condition. Treatments including pasteurization produced the highest accumulated release compared to the other treatments, even though the release under simulated gastrointestinal conditions was lower than for other treatments. The release in the simulated gastric phase was sustained but gradually increased in the simulated intestinal phase in all treatments (CA, CM, and CI). Furthermore, there was similar release behavior in SGF and SIF with or without digestion enzymes, indicating that these enzymes did not affect release from the CA, CM, and CI structures. Overall, the pH 3.0 treatment provided the lowest release and the highest retention of most of the antioxidant crude extracts from CSCG after the simulated gastric phase, which would be beneficial for bioavailability. The results suggested that the CM (alginate 2% *w/v* and maltodextrin 2% *w/v*) and CI (alginate 2% *w/v* and Inulin 5% *w/v*) compounds have good potential as carriers of antioxidant crude extracts from CSCG through a simulated gastrointestinal model. This system could deliver the phytochemical in the extract and release the compounds during digestion in the intestinal phase.

## Figures and Tables

**Figure 1 foods-12-01000-f001:**
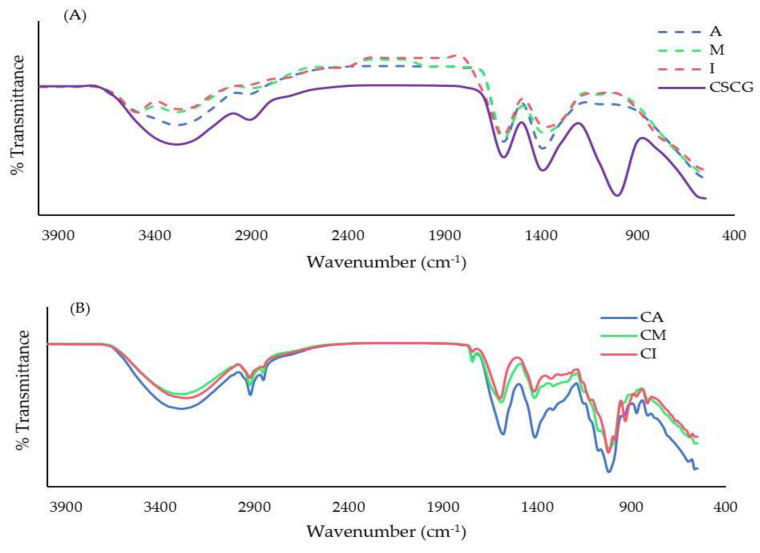
The FT−IR spectrum of antioxidant crude extract from CSCG (CSCG), A, M, and I (**A**), CA, CM, and CI (**B**).

**Figure 2 foods-12-01000-f002:**
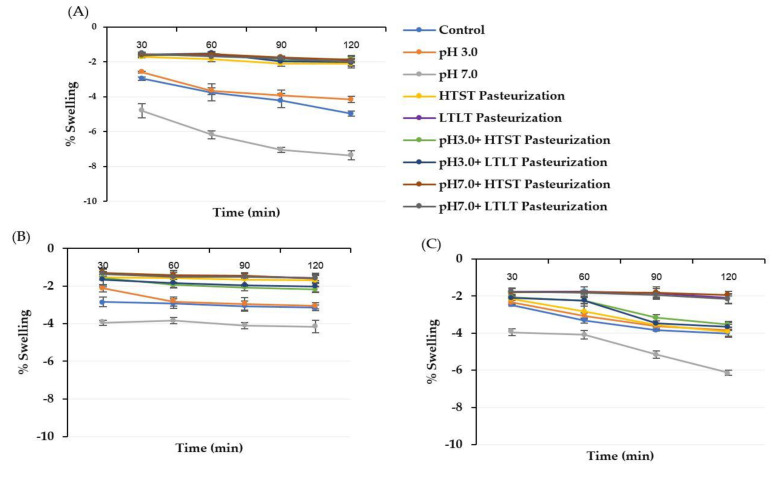
% Swelling of treated CA (**A**), CM (**B)**, and CI (**C)** hydrogel beads under SGF. The negative swelling percentage (−) indicates shrinkage behavior after the treatment. The error bars indicate the SD of the data.

**Figure 3 foods-12-01000-f003:**
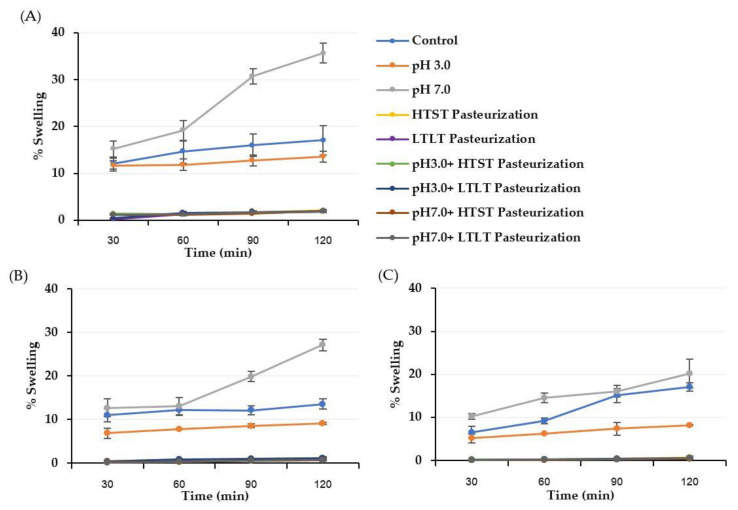
% Swelling of treated CA (**A**), CM (**B**), and CI (**C**) hydrogel beads under SIF. The positive swelling percentage indicates swelling behavior after the treatment. The error bars indicate the SD of the data.

**Figure 4 foods-12-01000-f004:**
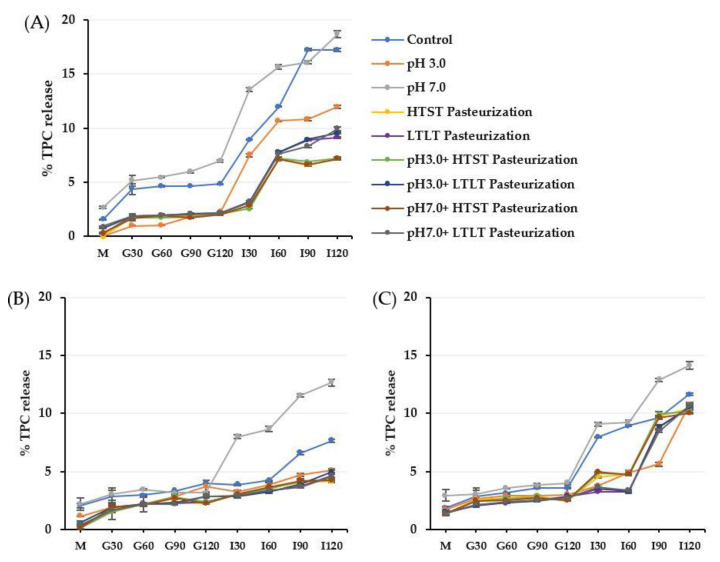
% TPC release of alginate (**A**), alginate/maltodextrin (**B**), and alginate/inulin (**C**) hydrogel beads in simulated gastrointestinal fluid. M = simulated oral phase, G30–G120 = simulated gastric phases at 30, 60, 90, and 120 min, respectively, I30–I120 = simulated intestinal phases at 30, 60, 90, and 120 min, respectively. The error bars indicate the standard deviation (SD) of the data.

**Figure 5 foods-12-01000-f005:**
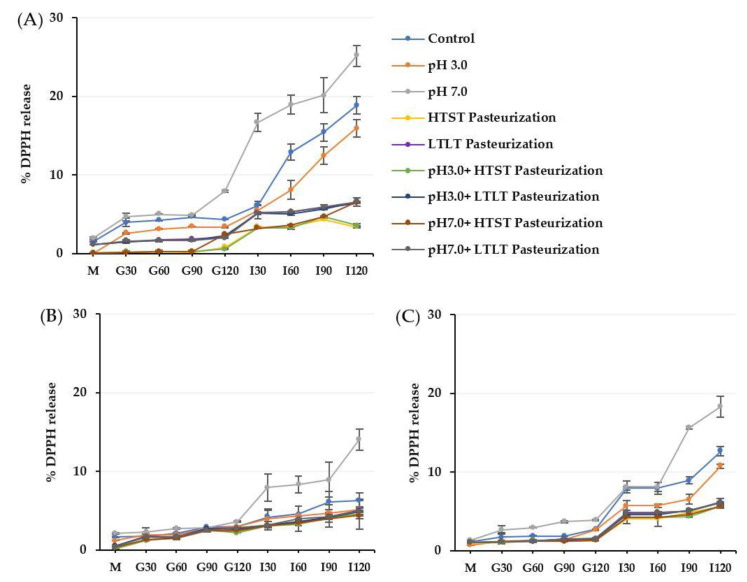
% DPPH release of alginate (**A**), alginate/maltodextrin (**B**), and alginate/inulin (**C**) hydrogel beads in simulated gastrointestinal fluid. M = simulated oral phase, G30–G120 = simulated gastric phases at 30, 60, 90, and 120 min, respectively, I30–I120 = simulated intestinal phases at 30, 60, 90, and 120 min, respectively. The error bars indicate the SD of the data.

**Figure 6 foods-12-01000-f006:**
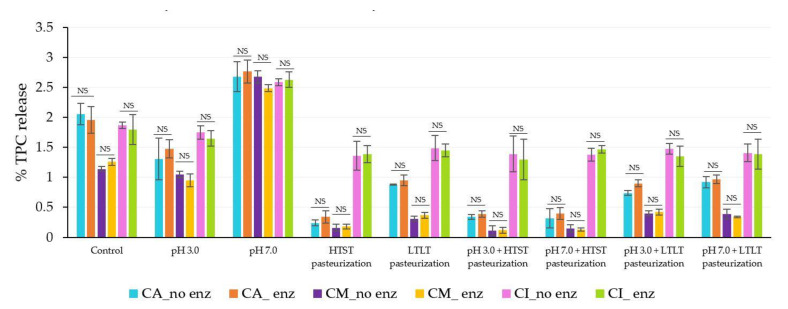
**%** TPC released during simulated gastrointestinal conditions in the oral phase with and without salivary amylase. NS indicated no significant differences at *p* ≤ 0.05. The error bars indicate the SD of the data. The control sample was microbeads after air-drying.

**Figure 7 foods-12-01000-f007:**
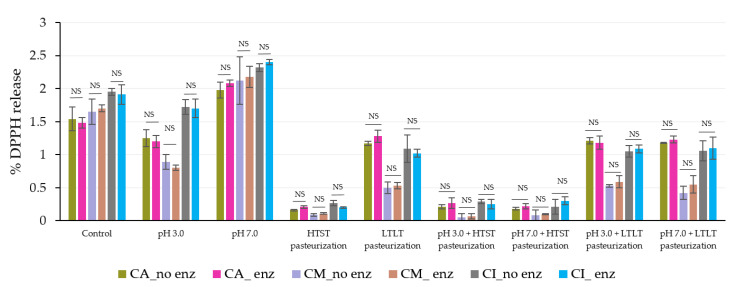
% DPPH released during simulated gastrointestinal conditions in the oral phase with and without salivary amylase. NS indicated no significant differences at *p* ≤ 0.05. The error bars indicate the SD of the data. The control sample was microbeads after air-drying.

**Figure 8 foods-12-01000-f008:**
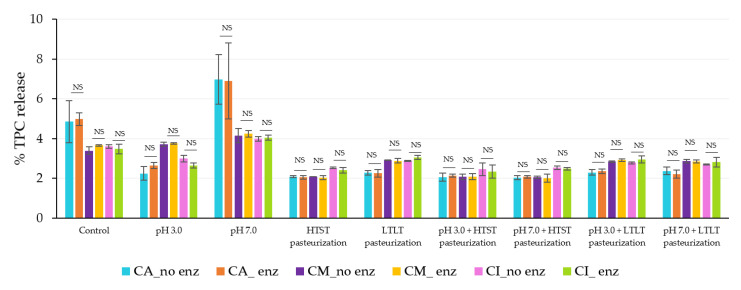
% TPC released during simulated gastrointestinal condition in the gastric phase (120 min) with and without pepsin. NS indicated no significant differences at *p* ≤ 0.05. The error bars indicate the SD of the data. The control sample was microbeads after air-drying.

**Figure 9 foods-12-01000-f009:**
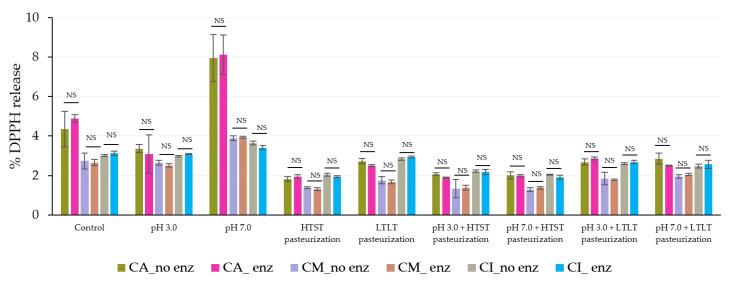
% DPPH release during simulated gastrointestinal conditions in the gastric phase (120 min) with and without pepsin. NS indicated no significant differences at *p* ≤ 0.05. The error bars indicate the SD of the data. The control sample was microbeads after air-drying.

**Figure 10 foods-12-01000-f010:**
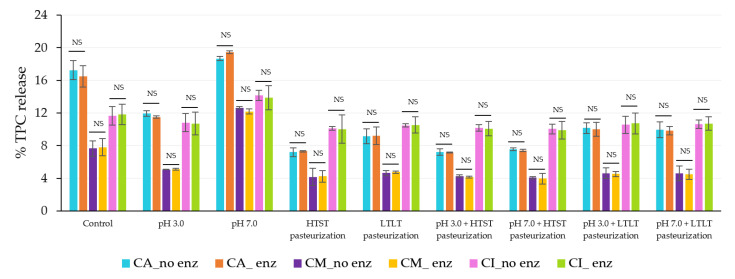
% TPC released during simulated gastrointestinal conditions in the intestinal phase (120 min) with and without pancreatic and bile salts. NS indicated no significant differences at *p* ≤ 0.05. The error bars indicate the SD of the data. The control sample was microbeads after air-drying.

**Figure 11 foods-12-01000-f011:**
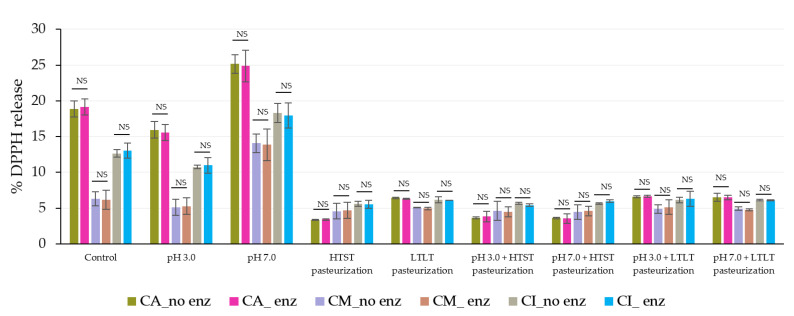
% DPPH released during simulated gastrointestinal conditions in the intestinal phase (120 min) with and without pancreatic and bile salts. NS indicated no significant differences at *p* ≤ 0.05. The error bars indicate the SD of the data. The control sample was microbeads after air-drying.

**Table 1 foods-12-01000-t001:** Ratio of antioxidant crude extract and encapsulated materials (alginate, alginate/maltodextrin, and alginate/inulin).

Antioxidant Crude Extract	Encapsulated Materials			Abbreviation
	Alginate (% *w/v*)	Maltodextrin (% *w/v*)	Inulin (% *w/v*)	
CSCG	2	-	-	CA
	2	2	-	CM
	2	-	5	CI

**Table 2 foods-12-01000-t002:** Average bead size of alginate (CA), alginate/maltodextrin (CM), and alginate/inulin (CI) hydrogel beads.

Encapsulated Sample	Size (mm)
CA	1.95 ^ns^ ± 0.97
CM	1.87 ^ns^ ± 0.66
CI	1.93 ^ns^ ± 0.74

^ns^ = Means ± standard deviation (SD) are not significantly different at *p* ≤ 0.05.

**Table 3 foods-12-01000-t003:** % Encapsulation efficiency of alginate (CA), alginate/maltodextrin (CM), and alginate/inulin (CI) hydrogel beads contained in encapsulated antioxidant crude extract from CSCG.

Encapsulated Sample	% Encapsulation Efficiency Based on TPC
CA	83.64 ^a^ ± 1.20
CM	89.76 ^c^ ± 1.90
CI	85.78 ^b^ ± 0.78

Means ± SD with different superscript letters indicate significant differences at *p* ≤ 0.05.

**Table 4 foods-12-01000-t004:** Swelling characteristics of alginate (CA), alginate/maltodextrin (CM), and alginate/inulin hydrogel beads contained in encapsulated antioxidant crude extract from CSCG.

Encapsulated Sample	% Swelling		
	CA	CM	CI
Control	ND	ND	ND
pH 3.0	−0.56 ^cA^ ± 0.03	−0.18 ^cB^ ± 0.02	−0.21 ^bB^ ± 0.05
pH 7.0	4.98 ^dB^ ± 0.10	3.11 ^dA^ ± 0.05	3.18 ^cA^ ± 0.15
HTST Pasteurization	−8.85 ^aA^ ± 0.05	−5.55 ^aB^ ± 0.03	−2.72 ^aC^ ± 0.21
LTLT Pasteurization	−6.19 ^bA^ ± 0.11	−3.01 ^bB^ ± 0.21	−2.16 ^aC^ ± 0.98
pH 3.0 + HTST Pasteurization	−8.46 ^aA^ ± 0.05	−5.88 ^aB^ ± 0.07	−2.54 ^aC^ ± 0.13
pH 7.0 + HTST Pasteurization	−8.58 ^aA^ ±0.06	−5.84 ^aB^ ± 0.03	−2.29 ^aC^ ± 0.01
pH 3.0 + LTLT Pasteurization	−6.58 ^bA^ ± 0.47	−2.98 ^bB^ ± 0.11	−2.01 ^aC^ ± 0.14
pH 7.0 + LTLT Pasteurization	−6.80 ^bA^ ± 0.04	−3.14 ^bB^ ± 0.35	−2.12 ^aC^ ± 0.57

^a–c^ Means ± SD within the columns with different lowercase superscripts indicate significant (*p* ≤ 0.05) differences. ^A–C^ Means ± SD within the rows with different uppercase superscripts indicate significant (*p* ≤ 0.05) differences. ND = not detected. The control sample was microbeads after air-drying without any treatment. The positive swelling percentage indicates a swelling behavior after treatment while a negative swelling percentage (-) indicates a shrinkage behavior after treatment.

**Table 5 foods-12-01000-t005:** % TPC release of alginate, alginate/maltodextrin, and alginate/inulin hydrogel beads containing encapsulated antioxidant crude extract from CSCG after passing through the simulated food process conditions.

Encapsulated Sample	% TPC Released		
	CA	CM	CI
Control	ND	ND	ND
pH 3.0	0.08 ^aA^ ± 0.04	0.02 ^aA^ ± 0.07	0.05 ^aA^ ± 0.06
pH 7.0	2.72 ^bC^ ± 0.13	1.95 ^bB^ ±0.05	1.18 ^bA^ ± 0.09
HTST Pasteurization	8.28 ^cB^ ± 0.08	6.43 ^cA^ ± 0.04	6.72 ^cA^ ± 0.15
LTLT Pasteurization	25.18 ^eB^ ± 0.26	20.95 ^dA^ ± 0.39	20.74 ^dA^ ± 0.47
pH 3.0 + HTST Pasteurization	8.89 ^dB^ ± 0.10	6.42 ^cA^ ± 0.02	6.54 ^cA^ ± 0.19
pH 7.0 + HTST Pasteurization	8.92 ^dB^ ± 0.08	6.58 ^cA^ ± 0.82	6.29 ^cA^ ± 0.11
pH 3.0 + LTLT Pasteurization	24.96 ^eB^ ± 0.57	20.81 ^dA^ ± 0.16	20.68 ^dA^ ± 0.51
pH 7.0 + LTLT Pasteurization	25.09 ^eB^ ± 0.32	19.97 ^dA^ ± 0.98	20.81 ^dA^ ± 0.99

^a–d^ Means ± SD within the columns with different lowercase superscripts indicate significant differences (*p* ≤ 0.05). ^A–C^ Means ± SD within the rows with different uppercase superscripts indicate significant differences (*p* ≤ 0.05). ND = not detected. The control sample was microbeads after air-drying.

**Table 6 foods-12-01000-t006:** % DPPH release of alginate, alginate/maltodextrin, and alginate/inulin hydrogel beads containing encapsulated antioxidant crude extract from CSCG after simulated food process conditions.

Encapsulated Samples	% DPPH Released		
	CA	CM	CI
Control	ND	ND	ND
pH 3.0	1.58 ^aC^ ± 0.11	1.09 ^aB^ ± 0.08	0.87 ^aA^ ± 0.01
pH 7.0	1.66 ^aB^ ± 0.08	1.25 ^aA^ ± 0.04	1.16 ^bA^ ± 0.05
HTST Pasteurization	5.91 ^bC^ ± 0.15	4.14 ^bB^ ± 0.23	3.85 ^cA^ ± 0.17
LTLT Pasteurization	11.95 ^cC^ ± 1.03	9.85 ^cA^ ± 0.64	10.15 ^dB^ ± 2.05
pH 3.0 + HTST Pasteurization	6.05 ^bB^ ± 0.94	3.87 ^bA^ ± 0.21	3.91 ^cA^ ± 0.55
pH 7.0 + HTST Pasteurization	5.98 ^bB^ ± 0.32	3.95 ^bA^ ± 0.67	4.05 ^cA^ ± 0.14
pH 3.0 + LTLT Pasteurization	11.84 ^cB^ ± 1.05	9.75 ^cA^ ± 1.22	9.94 ^dA^ ± 1.95
pH 7.0 + LTLT Pasteurization	12.00 ^cB^ ± 1.32	9.97 ^cA^ ± 0.98	10.07 ^dA^ ± 1.65

^a–c^ Means ± SD within the columns with different lowercase superscripts indicate significant differences (*p* ≤ 0.05). ^A–C^ Means ± SD within the rows with different uppercase superscripts indicate significant differences (*p* ≤ 0.05). ND = not detected. The control sample was microbeads after air-drying.

**Table 7 foods-12-01000-t007:** % TPC remaining of alginate, alginate/maltodextrin, and alginate/inulin hydrogel beads containing encapsulated antioxidant crude extract from CSCG after simulated food process conditions.

Encapsulated Sample	% TPC Remaining		
	CA	CM	CI
Control	99.85 ^dA^ ± 0.38	99.12 ^dA^ ± 0.38	99.45 ^cA^ ± 0.38
pH 3.0	98.45 ^dA^ ± 0.38	99.51 ^dB^ ± 0.18	99.28 ^cB^ ± 0.54
pH 7.0	96.41 ^cA^ ± 0.25	97.74 ^cB^ ± 0.26	97.65 ^cB^ ± 0.14
HTST Pasteurization	84.94 ^bA^ ± 0.14	90.28 ^bB^ ± 0.30	90.05 ^bB^ ± 0.28
LTLT Pasteurization	58.77 ^aA^ ± 1.21	65.68 ^aB^ ± 1.87	66.84 ^aB^ ± 1.95
pH 3.0 + HTST Pasteurization	85.02 ^bA^ ± 0.57	90.11 ^bB^ ± 0.27	90.20 ^bB^ ± 0.71
pH 7.0 + HTST Pasteurization	84.45 ^bA^ ± 0.65	89.92 ^bB^ ± 0.26	90.17 ^bB^ ± 1.25
pH 3.0 + LTLT Pasteurization	59.02 ^aA^ ± 1.38	65.77 ^aB^ ± 1.02	66.61 ^aB^ ± 0.64
pH 7.0 + LTLT Pasteurization	58.87 ^aA^ ± 0.99	65.48 ^aB^ ± 1.57	66.97 ^aB^ ± 0.84

^a–d^ Means ± SD within the columns with different lowercase superscripts indicate significant differences (*p* ≤ 0.05). ^A,B^ Means ± SD within the rows with different uppercase superscripts indicate significant differences (*p* ≤ 0.05). ND = not detected. The control sample was microbeads after air-drying.

**Table 8 foods-12-01000-t008:** % DPPH remaining of alginate, alginate/maltodextrin, and alginate/inulin hydrogel beads containing encapsulated antioxidant crude extract from CSCG after simulated food process conditions.

Encapsulated Sample	% DPPH Remaining		
	CA	CM	CI
Control	99.05 ^cA^ ± 0.85	99.61 ^cA^ ± 0.05	99.27 ^cA^ ± 0.67
pH 3.0	98.75 ^cA^ ± 2.05	99.05 ^cA^ ± 1.26	99.65 ^cA^ ± 0.51
pH 7.0	98.67 ^cA^ ± 1.95	99.17 ^cA^ ± 1.49	99.37 ^cA^ ± 1.98
HTST Pasteurization	90.15 ^bA^ ± 1.06	92.65 ^bB^ ± 1.06	93.65 ^bC^ ± 1.25
LTLT Pasteurization	75.80 ^aA^ ± 0.15	80.47 ^aB^ ± 1.11	81.47 ^aB^ ± 1.44
pH 3.0 + HTST Pasteurization	89.97 ^bA^ ± 1.47	92.18 ^bB^ ± 0.52	93.84 ^bC^ ± 0.65
pH 7.0 + HTST Pasteurization	90.22 ^bA^ ± 2.32	91.84 ^bA^ ± 1.36	93.55 ^bA^ ± 2.06
pH 3.0 + LTLT Pasteurization	75.17 ^aA^ ± 0.85	79.97 ^aB^ ± 1.65	80.95 ^aB^ ± 1.00
pH 7.0 + LTLT Pasteurization	75.46 ^aA^ ± 1.66	80.15 ^aB^ ± 1.20	81.67 ^aB^ ± 1.44

^a–c^ Means ± SD within the columns with different lowercase superscripts indicate significant differences (*p* ≤ 0.05). ^A–C^ Means ± SD within the rows with different uppercase superscripts indicate significant differences (*p* ≤ 0.05). ND = not detected. The control sample was microbeads after air-drying.

## Data Availability

Data is contained within the article.

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
