# Peer review of "Polyphenol Release and Antioxidant Activity of the Encapsulated Antioxidant Crude Extract from Cold Brew Spent Coffee Grounds under Simulated Food Processes and an In Vitro Static Gastrointestinal Model"

_foods, 2023, doi:10.3390/foods12051000_

Round 1

Reviewer 1 Report

In this manuscript, authors encapsulated the spent coffee ground extract containing phytochemicals using alginate, alginate/maltodextrin and alginate/inulin. Authors studied the stability of developed hydrogels at different processing conditions like pH 3, pH 7 and pasteurization conditions. The manuscript was written well with detailed analysis. Following are my queries and suggestions:

Ø  The detailed procedure for spent coffee crude extracted can be included in the materials and methods section.

Ø  Section 2.2 heading: Please correct as “Encapsulation efficiency”

Ø  Section 2.3: Author can give full form for LTLT and HTST pasteurization. Please cite the reference for the pasteurization procedure given in line 110 – 112.

Ø  Section 2.4: Authors can be a brief of TPC and DPPH procedure. It helps to readers to follow.

Ø  Table 4 footnote. I believe “a-f” means the statistical difference within the column and “A-C” represents statistical difference within the row. 

Author Response

Response to Reviewer

Reviewer#1

Comments and Suggestions for Authors

Dear Authors,

In this manuscript, authors encapsulated the spent coffee ground extract containing phytochemicals using alginate, alginate/maltodextrin and alginate/inulin. Authors studied the stability of developed hydrogels at different processing conditions like pH 3, pH 7 and pasteurization conditions. The manuscript was written well with detailed analysis.

Response Thank you very much for kindly reviewing our manuscript. Your comments and suggestions are value to us and help improve the quality of our manuscript. Thank you very much again.

Following are my queries and suggestions:

Ø  The detailed procedure for spent coffee crude extracted can be included in the materials and methods section.

Response Thank you very much for this suggestion. We’ve included the preparation details of the spent coffee ground extracts in the material and method section 2.1 (line 93-95). Thank you very much again.

Ø  Section 2.2 heading: Please correct as “Encapsulation efficiency”

Response Thank you very much for point out this spelling error. We already checked and corrected the word “Encapsulation efficiency” as shown in section 2.4 (Line 127), section 3.1 (Line 244) and section 3.1.3 (Line 275).

Ø  Section 2.3: Author can give full form for LTLT and HTST pasteurization. Please cite the reference for the pasteurization procedure given in line 110 – 112.

Response Thank you very much for your kind suggestion. We’ve already added the full term of LTLT and HTST in the abstract as the first-time add and then LTLT and HTST was chronologically used. The method of the LTLT and HTST pasteurization was added in the section 2.5 (Line 140-149). Thank you very much again.

Ø  Section 2.4: Authors can be a brief of TPC and DPPH procedure. It helps to readers to follow.

 Response Thank you very much for your suggestion. We’ve already added the TPC and DPPH procedure in the method section 2.4 (Line 134-137) and 2.6 (Line 182-188). Thank you very much again.

Ø  Table 4 footnote. I believe “a-f” means the statistical difference within the column and “A-C” represents statistical difference within the row. 

Response Thank you very much for your kind comments. We’ve already rechecked and corrected the footnote caption in all tables throughout our manuscript. Thank you very much again.

We would like to express our sincere thanks to you for spending your time reviewing our manuscript and if there is still any incorrect information, please do not hesitate to let us know.

Best Regards,

Tantawan Pirak, Corresponding Author

Reviewer 2 Report

The work with the title of “Antioxidant crude extract from cold brew spend coffee ground encapsulated with alginate, alginate/maltodextrin, and alginate/inulin: Release of polyphenols under simulate food processes and In-vitro static gastrointestinal model” studied the ionic gelation technique by alginate-calcium based encapsulation as a delivery matrix for antioxidant crude extracts from cold brew spent coffee grounds. Despite the fact that the paper is well written, it requires further revision in terms of the language and minor spelling errors. The title of the paper is too long. Please concise it.

As the study examines the release rate under the simulated GI tract condition, the role of crosslinking is obvious. Therefore, the authors need to further characterize all the samples with FTIR in order to compare the chemical structure of the beads they produced. Thus, they will be able to determine why swelling and, consequently, the release of polyphenols have been altered.

FTIR and SEM is a must for this study! Otherwise, there are works already in the literature similar to this study but different sources of polyphenols as the authors cited their works.

Why different acids were employed for providing different pH levels? 

For the statistical analysis please run visualization of the influence of categorical predictors on dependent variable as a response surface plot for more accuracy.

Fig. 1: why swelling percentage is negative? Please fix this since the negative percentage doesn’t sound scientific.

Line 145: why the equation is in the text? It should be Eq. 1 separately for the equations.

Author Response

Response to Reviewer

Reviewer#2

Comments and Suggestions for Authors

The work with the title of “Antioxidant crude extract from cold brew spend coffee ground encapsulated with alginate, alginate/maltodextrin, and alginate/inulin: Release of polyphenols under simulate food processes and In-vitro static gastrointestinal model” studied the ionic gelation technique by alginate-calcium based encapsulation as a delivery matrix for antioxidant crude extracts from cold brew spent coffee grounds. Despite the fact that the paper is well written, it requires further revision in terms of the language and minor spelling errors.

Response Thank you very much for kindly reviewing our manuscript. Your comments and suggestions are value to us and help improve the quality of our manuscript. Thank you very much again.

Following are my queries and suggestions:

- The title of the paper is too long. Please concise it.

Response Thank you very much for your suggestion. We’ve already concise the title as your suggestion to “Polyphenol release and antioxidant activity of the encapsulated antioxidant crude extract from cold brew coffee spent grounds under simulated food processes and in-vitro static gastrointestinal model”. Please kindly suggest us more if you think it should be changed further. Thank you very much again.

- As the study examines the release rate under the simulated GI tract condition, the role of crosslinking is obvious. Therefore, the authors need to further characterize all the samples with FTIR in order to compare the chemical structure of the beads they produced. Thus, they will be able to determine why swelling and, consequently, the release of polyphenols have been altered. FTIR and SEM is a must for this study! Otherwise, there are works already in the literature similar to this study but different sources of polyphenols as the authors cited their works.

Response  We very appreciated your considerate suggestion. Thank you very much for your guidance. The FTIR was performed, and the results was added to reveal the chemical structure of microbeads samples as shown in the result part in section 3.1.2 (Line 258-274).

- Why different acids were employed for providing different pH levels?

Response Thank you very much for your kind comment. Citric acid buffer and HCl were used in this investigation in the different objectives. The pH of the sample was changed to an acidic level using the citric buffer to simulate the acidic food pH. The pH of the in vitro digestion system was adjust using the HCl which is based on the method of INFOGEST. This standard method recommended the approach to increase the acidity during the in-vitro gastric phase. Hence, we decided to use different acid with different objectives. Thank you very much again for your kind suggestion.

- For the statistical analysis please run visualization of the influence of categorical predictors on dependent variable as a response surface plot for more accuracy.

Response Thank you for your valuable suggestion. We really appreciated it. We did not set the experimental design using RSM and the factors study was set up as a full factorial design in CRD to reveal the effect of the different encapsulated material on the polyphenol release during simulated gastrointestinal digestion and to monitor the changes of the release as affected by simulated thermal process and followed by in-vitro digestion. These information will be emphasized our understanding in designing the functional food and beverage in the later experiment which RSM might be a useful tools for this later experiment. Thank you very much again for your kind suggestions.

Fig. 1: why swelling percentage is negative? Please fix this since the negative percentage doesn’t sound scientific.

Response Thank you for kindly review our manuscript. We appreciated your comments and suggestions. According to our experiment, the shrinkage of the microbeads during the heat treatment was observed and resulted in a negative swelling percentage. We added the explanation of our observation in the Line 302-370 and all of the figures and tables that included the related result (Table 4 and Figure 2). Thank you very much again.

Line 145: why the equation is in the text? It should be Eq. 1 separately for the equations.

Response Thank you very much for your suggestion. We’ve already removed the equation from the text and re-written as equation 1 in section 2.4, equation 2 in section 2.5, equation 3-6 in section 2.6.1, equation 7-8 in section 2.6.2, and equation 9-10 in section 2.6.3. Thank you very much again.

We would like to express our sincere thanks to you for spending your time reviewing our manuscript and if there is still any incorrect information, please do not hesitate to let us know.

The certificate of grammatical correction from the native speaker was also attached below. Please kindly review our manuscript. Any further comments and suggestions are much appreciated. Thank you very much again.

Best Regards,

Tantawan Pirak, Corresponding Author

Reviewer 3 Report

Manuscript ID: foods-2183268

Title: Antioxidant crude extract from cold brew spend coffee ground encapsulated with alginate, alginate/maltodextrin, and alginate/inulin: Release of polyphenols under simulate food processes and In-vitro static gastrointestinal model

This manuscript is about the encapsulation of coffee extract by alginate based beads and their stability in static in vitro digestion model. The manuscript gives useful information, however, needs major revision according to the comments:

1.     Abstract: Please insert the ratio and concentration of encapsulating agent and extract in abstract. Also, it is better to use the main keywords.

2.     English should be improved. For instance, line 37, degrease or decrease, Line 472: physics chemical, etc.

3.     Introduction: Please explain more about maltodextrin and inulin properties which make them candidate as encapsulating agent.

4.     Materials and methods: Please insert the subtitle for preparation of crude extracted from CSCG. Also, authors should clarify all treatments (what was the Air dry sample in table 4?). The unites (w/w or v/v) in Table 1 should be inserted and cite the table number in line 87. Please insert more details in encapsulation efficacy and swelling properties. How the amount of TPC was determined? Also, the pH (line 123-129) and enzymes concentration should be inserted in in vitro release.

5.     The release and remain of TPC and antioxidant activity: Please insert more details about the concentrations used in this part. Moreover, DDPH was encapsulated in the same carriers?? Or antioxidant activity of encapsulated crude extract was measured by DPPH method? as the same, line 174, “antioxidant activity (DPPH)”. Please check and modify.

6.     This manuscript did not involve any chemical or morphological characterization. FTIR and SEM of beads should be inserted. Moreover, please insert the subtitle for microbeads size measurement in materials and methods.

7.     Line 207-208: “encapsulate efficiently was observed when increased maltodextrin concentration from 2 to 3% and inulin concentration from 5 to 10% (result not shown).” Please explain more details about concentrations in material and methods and related results.

8.     Table 3. Explain what you mean by significance in each column and row. Also, please recheck the statistical analysis of Table 4. How -0.56 (pH3) and 4.98 (pH 7) had no significant differences? Table 5 and 6: statistical letters recheck. Also, please correct the TPC% release in Table 6.

9.     Line 285: “Hence, the highest shrinkage and swelling were observed in the pH 7.0 treatment sample”. Shrinkage is opposite to the swelling process. This sentence is not correct. According to the figure 1, pH 7 had lower swelling (negative numbers exhibited the shrinkage of samples). Please recheck and cite the related figure and table.

10.  Line 525: higher thickness of hydrogel structure creates by maltodextrin and inulin” while the result of size was in contrast. Moreover, please explain how the size of CA was higher than CM and CI when using extra concentration of inulin or maltodextrin in beads?

11.  Please insert the statistical analysis letters in the figures 5 -10 and cite them in the appropriate place in the manuscript. 

12.  Authors should discuss the recent studies to show the importance of the present study as well as usage of maltodextrin and inulin. I suggest that discuss all the possibility to make the current presentation more interesting.

Author Response

Response to Reviewer

Reviewer#3

Comments and Suggestions for Authors

Title: Antioxidant crude extract from cold brew spend coffee ground encapsulated with alginate, alginate/maltodextrin, and alginate/inulin: Release of polyphenols under simulate food processes and In-vitro static gastrointestinal model

This manuscript is about the encapsulation of coffee extract by alginate based beads and their stability in static in vitro digestion model. The manuscript gives useful information, however, needs major revision according to the comments:

 Response Thank you very much for kindly reviewing our manuscript. Your comments and suggestions are value to us and help improve the quality of our manuscript. Thank you very much again.

  1. Abstract: Please insert the ratio and concentration of encapsulating agent and extract in abstract. Also, it is better to use the main keywords.

Response Thank you very much for your informative suggestions. We really appreciated and corrected as your suggestion. In the abstract, we’ve already included the ratio and concentration of encapsulating agent and extract (Line 13-17). The encapsulating agent as alginate, maltodextrin, and inulin were also included in the main key word. Thank you very much again.

  1. English should be improved. For instance, line 37, degrease or decrease, Line 472: physics chemical, etc.

Response Thank you very much for your kind comments. The improper typing was checked and corrected throughout the revised manuscript. Moreover, we’ve already submitted our manuscript for correcting the grammatical errors by the native speakers detailed at the end of this letter. If you think it should further correction, please do not hesitate to let us know. Thank you very much again.

  1. Introduction: Please explain more about maltodextrin and inulin properties which make them candidate as encapsulating agent.

Response Thank you for your valuable suggestion. More information on maltodextrin and inulin was already included in the introduction section (Line 69-77). Thank you very much again.  

  1. Materials and methods: Please insert the subtitle for preparation of crude extracted from CSCG. Also, authors should clarify all treatments (what was the Air dry sample in table 4?). The unites (w/w or v/v) in Table 1 should be inserted and cite the table number in line 87. Please insert more details in encapsulation efficacy and swelling properties. How the amount of TPC was determined? Also, the pH (line 123-129) and enzymes concentration should be inserted in in vitro release.

Response Thank you for your valuable suggestions. We’ve already included the spent coffee ground extract preparation in the method section 2.1 (Line 93-95). Moreover, the air dry sample explanation was added in the method in section 2.5 (Line 96-109) and referred as control sample in all of the table and figure footnote caption.

 The unit of weight/volume (w/v) was added to Table 1 and mentioned in method section 2.1.

The method in section 2.4 was now included with the information of the encapsulation efficiency and swelling properties (Line 128-138).

The TPC procedure was added in the method section 2.4 (Line 134-137).

Moreover, the in-vitro gastrointestinal procedure (method section 2.5.2 and 2.5.3) the pH, enzyme concentration, and bile salt concentration were already added according to your suggestions. Thank you very much again.

  1. The release and remain of TPC and antioxidant activity: Please insert more details about the concentrations used in this part.

Moreover, DDPH was encapsulated in the same carriers?? Or antioxidant activity of encapsulated crude extract was measured by DPPH method? as the same, line 174, “antioxidant activity (DPPH)”. Please check and modify.

Response Thank you very much for your kind suggestions. According to your suggestions, the concentration of the antioxidant crude extract from cold brew spent coffee ground was added in the method section 2.1 (Line 99-102). Moreover, we’ve already checked and rewritten to a better understanding in which the antioxidant activity of the encapsulated crude extract was measured by DPPH method (Line 182-187). Thank you very much again.

  1. This manuscript did not involve any chemical or morphological characterization. FTIR and SEM of beads should be inserted.

Moreover, please insert the subtitle for microbeads size measurement in materials and methods.

Response Thank you very much for your valuable suggestion. To reveal this characteristic, the FTIR was performed with the method described in the section 2.3 (Line 118-126). The FTIR analysis result was added in the results and discussion part (section 3.1.2, Line 258-274). Thank you very much again. Your comments help improve the merit of our study.  

  1. Line 207-208: “encapsulate efficiently was observed when increased maltodextrin concentration from 2 to 3% and inulin concentration from 5 to 10% (result not shown).” Please explain more details about concentrations in material and methods and related results.

Response Thank you for your kind comments. The concentration of each material was added and explained in the method section 2.1 (Line 99-102) and in the result section 3.1.3 (Line 275-301). Please let us know if there is still any further information needed. Thank you very much again.

  1. Table 3. Explain what you mean by significance in each column and row. Also, please recheck the statistical analysis of Table 4. How -0.56 (pH3) and 4.98 (pH 7) had no significant differences? Table 5 and 6: statistical letters recheck. Also, please correct the TPC% release in Table 6.

Response Thank you very much for your valuable comments. We apologize for our incorrect statistical letters. All the statistical data was rechecked and corrected as shown in Table 4-6. Thank you very much again.  

  1. Line 285: “Hence, the highest shrinkage and swelling were observed in the pH 7.0 treatment sample”. Shrinkage is opposite to the swelling process. This sentence is not correct. According to the figure 1, pH 7 had lower swelling (negative numbers exhibited the shrinkage of samples). Please recheck and cite the related figure and table.

Response Thank you very much for your valuable suggestions. We’ve already checked the results in Figure 2 and 3, and the sentence was rewritten as shown in Line 303-325. Thank you very much again.  

  1. Line 525: higher thickness of hydrogel structure creates by maltodextrin and inulin” while the result of size was in contrast. Moreover, please explain how the size of CA was higher than CM and CI when using extra concentration of inulin or maltodextrin in beads?

Response Thank you very much for your kind comments. The thickness is not significantly affected the size of the hydrogel beads (Table 2). The size of microbeads were control by the needle pore size then no significantly difference in size between the microbeads samples used in this study. To a better understanding, we removed this sentence (Line 525) and rewritten as shown in Line 604-611. Thank you very much again.

  1. Please insert the statistical analysis letters in the figures 5 -10 and cite them in the appropriate place in the manuscript. 

 Response Thank you very much for your kind suggestions. All the statistical analysis letters were rechecked, corrected, and cited in the appropriate section. Moreover, the letters indicated the statistically difference among treatment means were added in Figure 5-11. The indication of significant difference was also mentioned in the result. Thank you very much for your kind suggestion.

  1. Authors should discuss the recent studies to show the importance of the present study as well as usage of maltodextrin and inulin. I suggest that discuss all the possibility to make the current presentation more interesting.

Response  Thank you much for your kind suggestion. We’ve already discussed and added the usage of maltodextrin and inulin in the results and discussion (Line 303-326, and 423-435) and the conclusion parts (Line 704-709). Thank you very much again. Your suggestions help improve the originality of our manuscript.

We would like to express our sincere thanks to you for spending your time reviewing our manuscript and if there is still any incorrect information, please do not hesitate to let us know.

The certificate of grammatical correction from the native speaker was also attached below. Please kindly review our manuscript. Any further comments and suggestions are much appreciated. Thank you very much again.

Best Regards,

Tantawan Pirak, Corresponding Author

Round 2

Reviewer 2 Report

Dear Authors,

Thank you the revised version. Most of my concerns are now fixed.

Good luck

A.B

Author Response

Dear Reviewer,

        Thank you very much for spending your valuable time reviewing our manuscript. We're really appreciated. Thank you very much again.

Best Regards,

Tantawan Pirak, Corresponding Author

Reviewer 3 Report

Manuscript ID: foods-2183268

Title: Antioxidant crude extract from cold brew spend coffee ground encapsulated with alginate, alginate/maltodextrin, and alginate/inulin: Release of polyphenols under simulate food processes and In-vitro static gastrointestinal model

Dear Editor

This manuscript is modified well written. However, needs minor revision according to the comments:

1.     Materials and Methods:  New heading “2.1 Preparation of crude extract from cold brew coffee spent grounds” should be inserted.

2.     FTIR section: The FTIR spectrum of pure alginate, maltodextrin, inulin and crude extract should be inserted and compared with encapsulated forms (CA, CM, CI). The structure between the extract and encapsulating agents should be also revealed.

3.     Encapsulation efficiency section, line 282-287: “However, there was no significant difference in % encapsulation efficiently observed when the maltodextrin concentration increased from 2 to 3% (w/v) nor when the inulin concentration increased from 5 to 10% (w/v) (results not shown).” While according to the table 1 and method described in 2.1 section (line 99-102), maltodextrin at 2 % w/v and inulin at 5% w/v concentration were used. It should be mention if different concentrations of maltodextrin, alginate and inulin were used for preparation of beads.

Author Response

Response to Reviewer 3 (Round 2)

Reviewer’s comments

This manuscript is modified well written. However, needs minor revision according to the comments:

Response: Thank you very much for kindly revise our manuscript and give us the value comments and suggestions. Your contributions are much appreciated. We tried to improve the quality of our manuscript as best as we can, hence we analyzed more on FTIR of CA, AM and CI) as you suggested. The details were added into the revised version and the response point-by-point are as follow:

  1. Materials and Methods:  New heading “2.1 Preparation of crude extract from cold brew coffee spent grounds” should be inserted.

Response: Thank you very much, we really appreciate your suggestion. In response, the new heading Preparation of antioxidant crude extracts from CSCG was added to the manuscript's section 2.1.

  1. FTIR section: The FTIR spectrum of pure alginate, maltodextrin, inulin and crude extract should be inserted and compared with encapsulated forms (CA, CM, CI). The structure between the extract and encapsulating agents should be also revealed.

Response: Thank you very much for your kind suggestion. We’ve already analyzed the FT-IR spectra of pure alginate, maltodextrin, inulin and crude extract and added the information to the section of  materials and methods (Line 119-121) and results and discussion (Line 262-291). The structure of the extract and the encapsulated form was revealed. Please kindly review our revised manuscript and please let us know if there is other needed information. Thank you very much again for your valuable suggestion.

  1. Encapsulation efficiency section, line 282-287: “However, there was no significant difference in % encapsulation efficiently observed when the maltodextrin concentration increased from 2 to 3% (w/v) nor when the inulin concentration increased from 5 to 10% (w/v) (results not shown).” While according to the table 1 and method described in 2.1 section (line 99-102), maltodextrin at 2 % w/v and inulin at 5% w/v concentration were used. It should be mention if different concentrations of maltodextrin, alginate and inulin were used for preparation of beads.

Response: We very much appreciate your thoughtful evaluation. In section 3.1.3 on encapsulation efficiency, further information about the various concentrations utilized in the preliminary experiment was added (Line 299-306). Thank you very much again.
